# NO and GSH Alleviate the Inhibition of Low-Temperature Stress on Cowpea Seedlings

**DOI:** 10.3390/plants12061317

**Published:** 2023-03-14

**Authors:** Xueping Song, Zeping Xu, Jianwei Zhang, Le Liang, Jiachang Xiao, Zongxu Liang, Guofeng Yu, Bo Sun, Zhi Huang, Yi Tang, Yunsong Lai, Huanxiu Li

**Affiliations:** 1Department of Horticulturae, Sichuan Agricultural University, Chengdu 610000, China; 2Institute of Pomology and Olericulture, College of Horticulture, Sichuan Agricultural University, Chengdu 610000, China

**Keywords:** *Vigna unguiculata* (Linn.) *Walp.*, low-temperature nitric oxide glutathione

## Abstract

Low-temperature stress in early spring seriously affects the growth and development of cowpea seedlings. To study the alleviative effect of the exogenous substances nitric oxide (NO) and glutathione (GSH) on cowpea (*Vigna unguiculata* (Linn.) *Walp.*) seedlings under 8 °C low-temperature stress, 200 μmol·L^−1^ NO and 5 mmol·L^−1^ GSH were sprayed on cowpea seedlings whose second true leaf was about to unfold to enhance the tolerance of cowpea seedlings to low temperature. Spraying NO and GSH can eliminate excess superoxide radicals (O_2_^−^) and hydrogen peroxide (H_2_O_2_) to varying degrees, reduce the content of malondialdehyde and relative conductivity, delay the degradation of photosynthetic pigments, increase the content of osmotic regulating substances such as soluble sugar, soluble protein, and proline, and improve the activity of antioxidant enzymes such as superoxide dismutase, peroxidase, catalase, ascorbate peroxidase, dehydroascorbate reductase, and monodehydroascorbate reductase. This study revealed that the mixed use of NO and GSH played an important role in alleviating low temperature stress, and the effect of spraying NO alone was better than that of spraying GSH.

## 1. Introduction

Low temperature is one of the main factors affecting plant geographical distribution, and it will also affect the growth, development, and yield of crops, threatening crop production and food security [1]. Studies have shown that when plants are under low-temperature stress, the photosynthetic mechanism in the body is first inhibited, and the synthesis of photosynthetic pigments is blocked, which leads to a reduction in photosynthetic efficiency and further affects the growth and development of plants [2]. At the same time, plants have formed a series of physiological and metabolic activities, such as accumulation of osmotic regulatory substances, and activation of enzymatic and nonenzymatic systems, to alleviate the damage caused by low temperature to plants [3]. Osmoregulatory substances include soluble sugar (SS), soluble protein (SP), and proline (Pro) [4]. Low-temperature stress also produces excessive reactive oxygen species (ROS) in plants, and the accumulation of ROS damages biological macromolecules such as proteins, lipids, DNA, and cell membranes [5], leading to the peroxidation of lipid-unsaturated fatty acids in cell membranes, and promoting the degradation of polyunsaturated fatty acids to produce malondialdehyde (MDA), which is the oxidation product of unsaturated membrane lipids [6]. MDA content can be a key indicator to measure the degree of membrane lipid peroxidation in plants. The higher the MDA content, the higher the degree of plasma membrane peroxidation, and the weaker the ability of plants to withstand low temperatures [7].

Excessive ROS content in plant cells will destroy the ROS balance mechanism [8,9], but they will remove harmful H_2_O_2_ and O_2_^−^ from plants through the circulation of superoxide dismutase (SOD), peroxidase (POD), catalase (CAT) enzyme systems, and glutathione–ascorbic acid (GSH–AsA) [10]. Low-temperature stress on beans will affect the yield, quality, and metabolic rate of kidney beans [11]. There are two stages of sensitivity to low temperature: the first is the seedling stage [12] and the second is the flowering stage. When the temperature at night is lower than 8 °C, it will inhibit the formation of pods [13]. As exogenous substances, NO and GSH can improve the physiological state and antioxidant system of plants under abiotic stress conditions such as salt, drought, low temperature, and heavy metals, to enhance their stress tolerance [14,15].

Nitric oxide (NO) is a free radical that gives rise to a family of derived molecules designated as reactive nitrogen species (RNS), such as nitrogen dioxide (NO_2_), peroxynitrite anion (ONOO^−^), or S-nitroso thiols (SNOs) [16]. In plants, NO functions as a signaling molecule that is involved in a variety of physiological processes, including seed germination, root growth, stomatal closure, senescence, and response to abiotic and biotic stresses [17]. Studies have shown that spraying NO on peaches before low-temperature storage can delay the increase in MDA content, enhance the activity and content of related enzymes in the GSH–ASA cycle, and improve their low-temperature tolerance [18]. Spraying GSH can promote the expression level of GSH–ASA cycle-related enzyme genes in sweet pepper fruits before cold storage [19] and restore the growth of rice seedling roots under low-temperature stress [20]. Sodium nitroso ferricyanide (SNP), whose chemical formula is Na_2_[Fe (CN)_5_NO], is used as a direct donor of NO and can release it when dissolved in water [21].

The cowpea (*Vigna unguiculata* (Linn.) *Walp.*) is an annual twine, grassy vine, or nearly erect herb of the genus Cowpea [22], which ranks fifth worldwide as a source of plant protein and fiber [23]. As exogenous substances, the synergistic effect of GSH and NO have certain alleviating effects on other food crops and vegetables under various abiotic conditions, but the mechanism of alleviating cowpea growth under low temperature remains unclear. This study provides a theoretical basis for improving the cold tolerance of cowpea seedlings by researching their physiological characteristics by spraying NO and GSH alone and in combination, and has a certain significance for the sustainable development of the cowpea industry.

## 2. Results

### 2.1. Effects of Low-Temperature Stress on the Phenotype and Physiological Indices of Cowpea Seedlings

In this study, cowpea seedlings began to wilt on the first and second days of low-temperature stress and completely wilted on the third day. However, the seedlings sprayed with exogenous NO and GSH had almost no change in the first two days. On the third day, seedlings sprayed with NO or GSH alone wilted, while those sprayed with NO+GSH had little change (Figure 1A).

Spraying 200 μmol/L NO and 5 mmol/L GSH on the leaves can reduce the relative conductivity (REC) and MDA content of seedlings to varying degrees. Under the T3 treatment, the REC content of cowpea was significantly reduced by 32.48% compared with CK on the first day, and the MDA content was significantly reduced by 44.71% compared with CK on the third day. In the T1 treatment, the REC and MDA content of cowpea on the second day decreased by 17.43% and 32.48%, respectively, compared with CK (Figure 1B,C), which reached a significant level, while the effect of the T2 treatment at the same time had the same trend (*p* < 0.05).

### 2.2. Superoxide Anion (O_2_^−^) and H_2_O_2_ Contents

Under low-temperature stress, the production rate of superoxide anion (O_2_^−^) and the content of H_2_O_2_ in the leaves of seedlings increased with the stress duration. Compared with CK, spraying NO and GSH could significantly alleviate the increase in the O_2_^−^ production rate and H_2_O_2_ content in leaves. On the first, second, and third days of stress, the production rate of superoxide anions in cowpea seedlings treated with T3 was significantly reduced by 28.77%, 20.00%, and 21.46% compared with CK, and the production rate of O_2_^−^ was the lowest among the four treatments. Compared with CK, the H_2_O_2_ content decreased by 26.98%, 32.40%, and 31.48%. Among the four treatments, the content of H_2_O_2_ in the T3 treatment was the lowest. Compared with CK, the production rates of superoxide anions and H_2_O_2_ content in the T1 and T2 treatments were significantly lower than those in CK at each stress stage, and the mitigation effect of T2 was more obvious than that of T1. (Figure 2) (*p* < 0.05).

### 2.3. The Content of Osmotic Adjustment Substance

The contents of SS, Pro, and SP in cowpea seedlings increase with the extension of low-temperature stress time. Compared with CK, the T3 treatment can significantly increase the contents of SS, Pro, and SP in leaves. In the T3 treatment, the SS content increased by 103.49% on the first day, the Pro content increased by 36.37%, which was significantly higher than that in CK, and the SP content reached its peak on the third day, which was significantly increased by 8.50% compared with that in CK (Figure 3). (*p* < 0.05).

### 2.4. The Content of Photosynthetic Pigment

After low-temperature stress, the content of photosynthetic pigments (chlorophyll a, chlorophyll b, total chlorophyll, and carotenoids) in cowpea seedlings decreased with the extension of stress time. However, spraying exogenous substances NO and GSH in different treatments significantly delayed the degradation of photosynthetic pigments, and the effect of the T3 treatment was the most obvious. The content of photosynthetic pigment of cowpea was 47.93%, 48.65%, 48.15%, and 32.35% higher than that of CK on the third day of T3 treatment. The alleviative effect of each treatment on the leaves of white cowpea seedlings was T3 > T1 > T2 > CK, and in the whole stress process, each treatment reached a significant level compared with CK (Figure 4) (*p* < 0.05).

### 2.5. Activity Changes of SOD, POD and CAT

With the passage of low-temperature time, SOD activity increased, POD activity increased first and then decreased, and CAT activity decreased first and then increased. Spraying exogenous NO and GSH could improve the antioxidant enzyme activity of cowpea during cold-stress treatment. In the whole treatment process, the three oxidase activities of cowpea in the T3 treatment were significantly improved compared with CK to varying degrees, but the SOD and POD enzyme activities reached the peak value on the first day, increasing by 129.15% and 81.89%, respectively, while CAT increased by 54.05% on the second day (Figure 5) (*p* < 0.05).

With the extension of low-temperature stress time, spraying exogenous NO and GSH can improve the antioxidant enzyme activities of cowpea, but the changes of different enzyme activities are inconsistent. SOD and POD activities gradually increased, and CAT activity first decreased and then increased. T1 and T2 treatments also reached a significant level in multiple treatment periods compared with CK, and showed the same trend of change (Figure 5).

### 2.6. Changes in Antioxidant Enzyme Activity in the GSH–ASA Cycle

During the low temperature, the activities of DHAR and MDHAR of cowpea seedlings first increased and then decreased, and the activities of APX gradually increased. NO and GSH effectively increased the enzyme activities of cowpea seedlings. The APX activity of cowpea under the T3 treatment increased by 40.17%, 42.62%, and 48.54%, the DHAR activity increased by 25.24%, 48.41%, and 27.09%, and the MDHAR activity increased by 25.41%, 21.30%, and 30.71%, respectively, compared with CK on the first, second, and third days of stress treatment (Figure 6) (*p* < 0.05).

### 2.7. Pearson Correlation Analysis

In the experiment, we found that when seedlings were subjected to low-temperature stress, their relative electrical conductivity and other physiological indicators showed a certain correlation. In order to further intuitively display the relationship between each indicator, correlation analysis was conducted on the data based on heat map, as shown in Figure 7. The relative conductivity was significantly correlated with Pro, SOD, POD, CAT, APX, DHAR, and MDA (*p* < 0.05, *p* < 0.01). The relative conductivity was negatively correlated with Pro, SOD, POD, CAT, APX, and DHAR, but positively correlated with MDA. There was no significant difference between the relative electrical conductivity and superoxide anion, hydrogen peroxide, soluble sugar, total chlorophyll, chlorophyll a, chlorophyll b, and carotenoid (*p* > 0.05).

## 3. Discussion

Low temperatures have many effects on plants. In this study, cowpea seedlings began to wilt on the first and second days of low-temperature stress, and completely wilted on the third day. However, the seedlings sprayed with exogenous NO and GSH had almost no change in the first two days. On the third day, seedlings sprayed with NO or GSH alone wilted, while those sprayed with NO+GSH had little change (Figure 1A). In addition, through the measurement of physiological indicators O_2_^−^, H_2_O_2_, MDA, and REC of seedling leaves, it was found that with the extension of low-temperature stress, excessive O_2_^−^ and H_2_O_2_ accumulated (Figure 2), leading to an imbalance in the ROS metabolism, intensifying the lipid peroxidation of the cell membrane, causing cell membrane damage, and inducing an increase in MDA content and REC (Figure 1B,C), which was consistent with the results of previous studies on maize melatonin spray to alleviate low temperature stress [24]. Compared with the low-temperature control, exogenous NO, GSH, and NO+GSH can significantly reduce O_2_^−^ production rate and H_2_O_2_ (Figure 2), MDA, and REC content (Figure 1A), which can effectively alleviate the stress of low temperatures on cowpea seedlings. This is consistent with the research results of tomato seedlings [25] and eggplant [26] under low temperatures. Some studies suggest that NO may act on H_2_O_2_ in mediating stress tolerance [27] (Figure 8).

When plants are under low-temperature stress, the osmotic potential of cells and water loss can be reduced and biological macromolecules can be protected by increasing the content of osmotic regulators in cells. In this study, spraying exogenous substances NO and GSH induced the content of SS, SP, and Pro in seedling leaves to increase significantly compared with the control, and the effect of NO+GSH was the best (Figure 3), whereby the increase in SS and Pro content was more obvious than the change of SP content. This result shows that exogenous NO+GSH may promote the increase of endogenous NO levels in cowpea seedlings when resisting low-temperature stress, and alleviate electrolyte leakage and lipid peroxidation caused by low-temperature stress by increasing soluble sugar, proline, and GSH. This allows cells to maintain low osmotic potential and enhances the cold tolerance of plants under low-temperature stress. This is consistent with previous studies on cold-stressed rice seedlings [28], tea plants [29], and salt-stressed spinach [30].

In the growth stage of plant seedlings, low temperatures cause complex symptoms of cell dysfunction, such as wilting, growth inhibition, yellowing, and even death [31]. At the same time, it will destroy the structure and function of the photosynthetic system, affect photosynthesis, and degrade photosynthetic pigments, thus affecting the synthesis of organic substances in the Calvin cycle. In this study, with the extension of stress time, total chlorophyll, chlorophyll a, chlorophyll b, and carotenoids were degraded continuously, but spraying NO and GSH effectively alleviated the degradation of various photosynthetic pigments (Figure 4), and the treatment of NO+GSH was the most significant, which shows that NO and GSH can protect the photosynthetic system of cowpea seedlings. Some studies have shown that the reduction of photosynthetic capacity may be related to the reduction in key enzymes in the Calvin cycle, such as sedoheptulose-1,7-bisphosphatase and fructose 1,6-bisphosphatase [32], while Jiang et al. [33] also reported that the expression of Calvin cycle enzyme genes is positively related to the GSH–GSSG ratio, and the redox state plays an important role in the stability of Calvin cycle enzymes. Therefore, NO and GSH may alleviate the damage to cowpea seedlings caused by low-temperature stress by controlling the expression of key enzyme genes in the light response and Calvin cycle [31].

This study found that, with the extension of stress days, six antioxidant enzyme indicators had different change trends. The activities of SOD, POD, CAT, and APX increased significantly, and the activities of DHAR and MDHAR increased first and then decreased (Figure 5 and Figure 6), which proved that plants could increase their tolerance to low temperatures through the antioxidant enzyme system. Studies have shown that NO can reduce the accumulation of ROS by activating the activity of antioxidant enzymes. At the same time, NO can also enhance the production of antioxidant enzymes through the MAPK pathway [34], and can also enhance the accumulation of antioxidant enzymes through the S-nitroso (SNO) process [35]. As a strong oxidant, GSH can directly remove ROS in cells (Figure 8), and detoxify ROS by assisting GSH–ASA circulation [36]. In this study, the spraying of exogenous NO and GSH can significantly enhance the activities of SOD, POD, CAT, APX, DHAR, and MDHAR in cowpea seedlings, and the effect of NO+GSH treatment was the best (Figure 5 and Figure 6), indicating that NO and GSH could synergically alleviate the stress of low temperature on cowpea seedlings, which is consistent with the research results of Yao et al. [20].

Correlation analysis showed that relative conductivity was highly significantly negatively correlated (*p* < 0.01) with proline, superoxide dismutase, peroxidase, catalase, ascorbic acid peroxidase, and dehydroascorbic acid reductase, but it was positively correlated (*p* < 0.01) with the malondialdehyde (Figure 7). The results showed that REC and MDA could measure the damage to seedlings caused by low-temperature stress to a certain extent, and various antioxidant enzymes could resist the damage of low-temperature stress (Figure 5 and Figure 6). In this study, spraying GSH+NO significantly increased the activities of various antioxidant enzymes and increased the low-temperature tolerance of seedlings. NO regulates carbohydrate metabolism at the posttranslational level and glutathione (GSH) and methionine metabolism at the transcriptional level [37]. NO interacts with GSH to form S-nitroso glutathione (GSNO), which can act as an NO reservoir, mediate the regulation of intracellular NO level and protein nitrosation processes, directly or indirectly affect all physiological processes in plants, and participate in the defense mechanism against stress [38,39]. At the same time, as signal molecules, NO and GSH have crosstalk effects with other signals to enhance the adaptability of plants to adversity [39]. Previous studies have shown that the synergistic effect of NO and GSH can enhance the better resistance of plants [24]. In this study, the mixed spray of NO and GSH had the best effect, which is consistent with previous studies. However, the effect of NO in a single-spraying treatment is better than that of GSH, and its mechanism needs to be further explored.

## 4. Materials and Methods

### 4.1. Materials and Exogenous Reagents

The cowpea “SanchiBaixue” is grown in the laboratory provided by the Institute of Pomology and Olericulture of Sichuan Agricultural University. NO donor sodium nitroprusside (SNP) were purchased from Chengdu Cologne Chemicals Limited Company, and reduced L-glutathione (GSH) was purchased from Beijing Solebo Technology Limited Company, and the two reagents were prepared at the same time for use.

### 4.2. Experimental Materials and Treatment

The experiment was carried out on the Chengdu Campus of Sichuan Agricultural University from March to June 2022. Healthy and plump seeds were wrapped in mesh bags and placed in a beaker, soaked in warm water at 55 °C for 30 min, soaked at 25 °C for 6 h, then wrapped with gauze. The seeds were placed in a dark incubator at 30 °C for constant temperature germination. After 80% of the seeds grew white buds, seeds with the same germination growth were selected and sown on 32 deep-hole seedling trays (5 cm × 15 cm × 5 cm), with 6 seedling trays for each variety and 4 seeds for each well. The seedling tray was placed in an outdoor suitable growth environment, and the nutrient soil moisture was kept at 60%. After 12–13 days, when the first pair of true leaves had fully unfolded (based on the growth of the second pair of true leaves), we selected seedlings with good growth and consistent development statuses for exogenous treatment. Before low-temperature treatment, cowpea seedlings were placed in an artificial intelligence climate box for preculture, whose conditions included a temperature of 25/18 °C, photoperiod 12 h/12 h (day and night), light intensity 300 μmol m^−2^ s^−1^, and relative humidity 75%. After 24 h of preculture, the low-temperature treatment was carried out at a temperature of 8/8 °C, a light intensity 300 μmol m^−2^ s^−1^, and a relative humidity of 75%. The experiment was divided into four treatments, namely clear water control (CK), 200 μmol/L NO (T1), 5 mmol/L GSH (T2), and 200 μmol/L NO + 5 mmol/L GSH (T3), until water drops were formed on the leaf surface without dropping. There were 16 seedlings in each treatment, and each treatment was repeated three times and simultaneously. Cowpea seedlings were randomly placed in an intelligent artificial incubator. The true leaves of two seedlings were cut on the first day, second day, and third day under low-temperature stress, and random sampling was conducted according to the sample size required for each index. The samples were stored at −80 °C for subsequent determination. Three biological replicates were conducted for each variety (line) and the results were averaged.

### 4.3. Physiological and Biochemical Index Measurement

#### 4.3.1. MDA and Relative Conductivity (REC) Measurement

The MDA content was measured according to the method of Cakmak and Horst [40] with minor modifications. In brief, 0.5 g fresh leaf samples were homogenized at 4 °C with 10% (*w*/*v*) trichloroacetic acid (TCA). After centrifugation, the supernatant was incubated with an equal volume of 0.5% thiobarbituric acid (TBA) at 100 °C for 30 min, and then the absorbance was measured at 450, 532, and 600 nm.

A 0.1 g blade punched with a 0.5 cm diameter punch was weighed, placed into a 50 mL centrifuge tube, added to 30 mL deionized water, and placed on a shaker at 150 rpm to shade and shake for 6 h. The conductivities of deionized water (S_0_) and immersion solution (S_1_) were measured with a conductance meter. The immersion solution was boiled in a water bath for 30 min and cooled to room temperature before measuring the conductivity of the solution (S_2_). The relative conductivity is calculated as follows: relative conductivity (%) = (S_1_ − S_0_) × 100/ (S_2_ − S_0_).

#### 4.3.2. Determination of Superoxide Anion (O_2_^−^) and H_2_O_2_ Content

The O_2_^−^ production rate was measured by analyzing nitrite formation from hydroxylamine in the presence of O_2_^−^ [41]. Each 0.5 g of frozen leaf segment was homogenized with 1 mL of 50 mM phosphate buffer (pH 7.8) and centrifuged at 12,000× *g* for 10 min. The incubation mixture contained 1 mL of 1 mL of 50 mM phosphate buffer (pH 7.8). Hydroxylamine hydrochloride is 1 mL of 1 mM hydroxylamine and 1 mL of the supernatant; after incubation at 25 °C for 20 min, 17 mmol/L sulfanilamide and 7 mmol/L α-naphthylamine were added to the incubation mixture. Absorbance in the concentration solution was read at 530 nm.

Hydrogen peroxide levels were determined according to Sergiev et al. [42]. Leaf tissues (0.5 g) were homogenized in an ice bath with 5 mL 0.1% (*w*/*v*) TCA. The homogenate was centrifuged at 12,000× *g* for 15 min, and 0.5 mL of the supernatant was added to 0.5 mL of 10 mM potassium phosphate buffer (pH 7.0) and 1 mL of 1 M KI. The absorbance of the supernatant was read at 390 nm. The content of H_2_O_2_ was given on a standard curve.

#### 4.3.3. Osmotic Substance Content Measurement

The content of soluble sugar (SS) was determined by the anthrone colorimetric method [43], and the content of proline (Pro) was determined by the sulfosalicylic acid method [44]. The soluble protein content (SP) was determined by Coomassie brilliant blue G-250 staining [45].

#### 4.3.4. Photosynthetic Pigment Assay

The leaves of 0.2 g were weighed and determined by the immersion method (ethanol:acetone = 1:1, *v*:*v*), and the contents of chlorophyll a, chlorophyll b, carotenoids, and total chlorophyll were calculated [46]

#### 4.3.5. Assay of Superoxide Dismutase (SOD), Peroxidase (POD), and Catalase (CAT) Activity

A total of 1 g of leaves was ground on ice by adding precooled 50 mmol/L phosphate (PBS) buffer (pH 7.8) containing 1 g polyvinyl pyrrolidone (PVP), 2 mmol/L dithiothreitol and 0.1 mmol/L ethylenediaminetetraacetic acid (EDTA-NA_2_), and homogenized to 8 mL The samples were centrifuged at a low temperature for 10 min at 10,000× *g*, and the supernatant was taken for subsequent enzyme activity determination. SOD activity was determined by the azote–tetrazole photochemical reduction method, POD activity was measured by guaiacol method, and CAT activity was determined by the UV absorption method [43].

#### 4.3.6. Assay of Enzyme Activity in Glutathione–Ascorbate Cycle (GSH–ASA Cycle)

A total of 0.5 g of frozen leaves under different treatments in each period was added to a precooled PBS buffer (2% PVP, 2 mM ASA, 5 mM EDTA-NA_2_, pH 7.8). These were ground to homogenate under an ice bath with a constant volume of 10 mL. The samples were centrifuged at 12,000× *g* for 20 min at 4 °C. The supernatant was prepared for the determination of related enzyme activities.

The activity of APX was determined according to the method of Nakano and Asada [47], which was slightly modified. The mixed=reaction solution (0.1 mM EDTA-Na_2_, ASA 5 mM, H_2_O_2_ 20 mM) was added into a 0.1 mL enzyme solution, and the absorbance was measured at 290 nm wavelength for 10 s/time for 1 min. The kinetic change of 20 s was used to calculate the reaction rate.

The activities of DHAR and MDHAR were determined according to the method of Duan et al. [30], which was slightly modified. The reaction solution (0.1 mM EDTA-NA_2_ + 70 mM GSH + 8 mM ASA) in advance ice bath was absorbed, and the absorbance was measured at 265 nm wavelength for 10 s for 1 min. The reaction rate was calculated by taking the kinetic change of over 20 s.

### 4.4. Data Analysis and Processing

Data were plotted by Microsoft Excel 2010 and processed by SPSS 22.0 software, and Duncan’s significance analysis was performed (*p* ≤ 0.05). Data in the figures are the mean ± standard deviation.

## 5. Conclusions

Under low-temperature stress, spraying NO and GSH can increase the chlorophyll content and improve the absorption and conversion of light energy, which is beneficial to the photosynthesis of cowpea seedlings. In addition, spraying NO and GSH can promote the activity of antioxidant enzymes in leaves, increase the contents of osmotic regulatory substances, decrease the contents of REC and MDA in leaves to maintain the integrity of leaf cell structure, and improve the tolerance of cowpea seedlings to low temperature. At the same time, the mixed spraying of NO and GSH had a better alleviating effect on low-temperature stress than the single spraying of NO or GSH.

## Figures and Tables

**Figure 1 plants-12-01317-f001:**
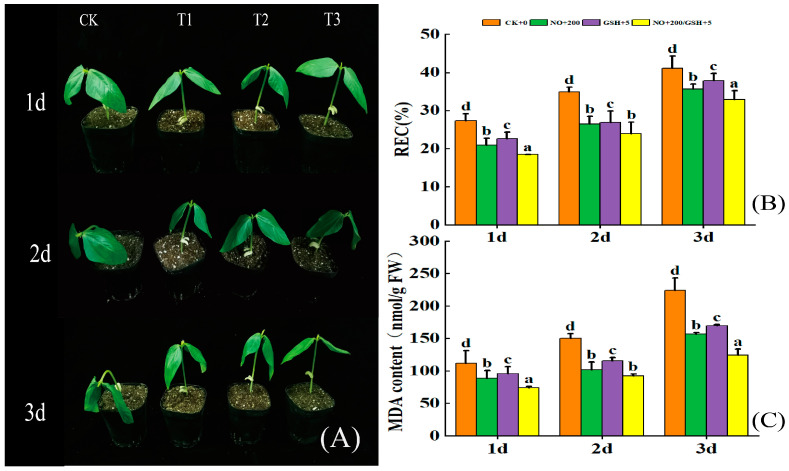
Effects of low-temperature stress on phenotype (**A**) (CK: low-temperature clear-water control, T1: 200 μmol/L NO, T2: 5 mmol/L GSH, T3: 200 μmol/L NO + 5 mmol/L GSH) and physiological indices (**B**,**C**) of cowpea seedlings. Different letters indicate statistically significant differences according to Duncan’s multiple range test *p* ≤ 0.05.

**Figure 2 plants-12-01317-f002:**
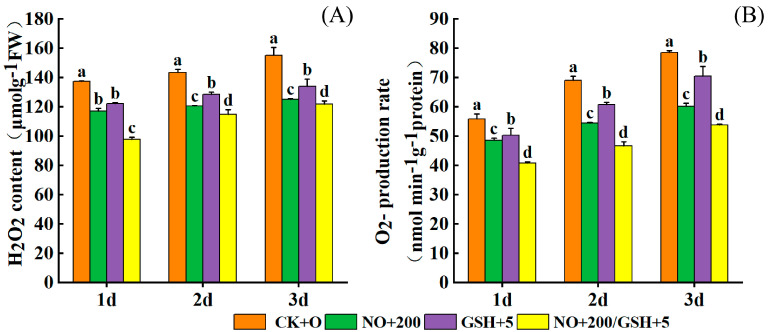
Effects of NO and GSH on H_2_O_2_ (**A**) and O_2_^−^ (**B**) of cowpea seedlings under low-temperature stress. Each value is presented as the mean ± standard error (n = 3). The differences among the treatments indicated with the same letter vertically were not significant according to Duncan’s multiple range test at *p* < 0.05.

**Figure 3 plants-12-01317-f003:**
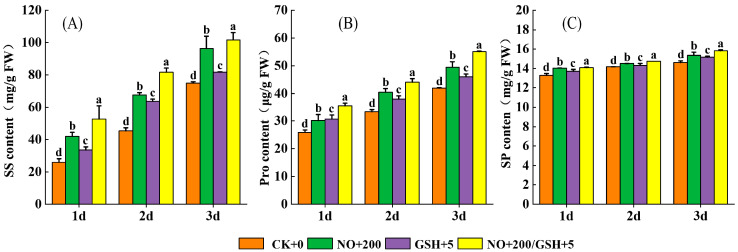
Effects of NO and GSH on ((**A**): SS), ((**B**): Pro), and ((**C**): SP) of cowpea seedlings under low-temperature stress. Different letters indicate statistically significant differences according to Duncan’s multiple range test at *p* < 0.05.

**Figure 4 plants-12-01317-f004:**
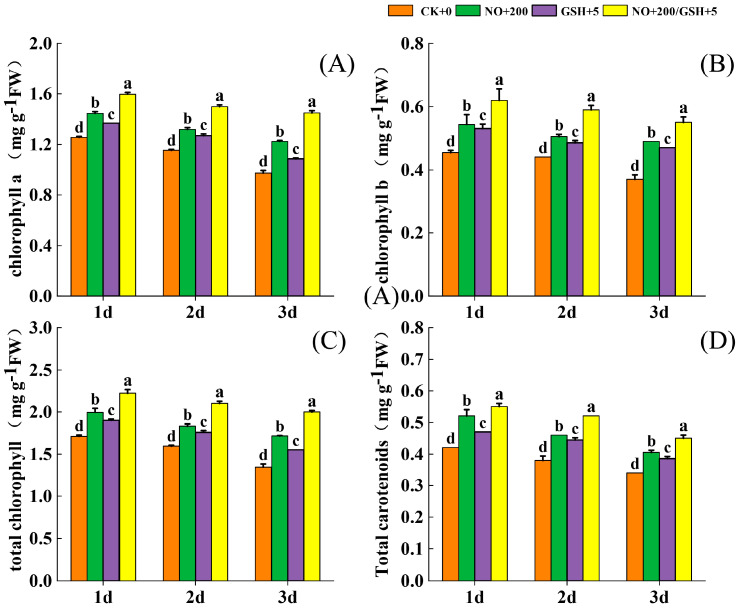
Effects of NO and GSH on photosynthetic pigment ((**A**): chlorophyll a, (**B**): chlorophyll b, (**C**): total chlorophyll, (**D**): carotenoid) contents of cowpea seedlings under low-temperature stress. Different letters indicate statistically significant differences according to Duncan’s multiple range test at *p* < 0.05.

**Figure 5 plants-12-01317-f005:**
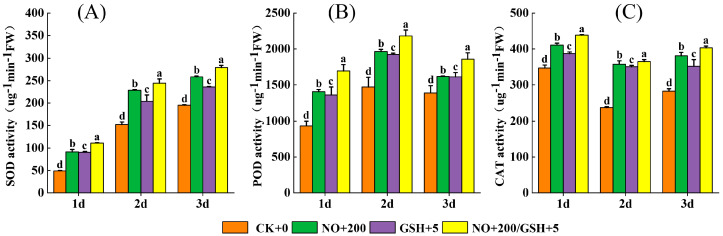
Effects of NO and GSH on the activities of ((**A**): SOD), ((**B**): POD), and ((**C**): CAT) of cowpea seedlings under low-temperature stress. Different letters indicate statistically significant differences according to Duncan’s multiple range test at *p* < 0.05.

**Figure 6 plants-12-01317-f006:**
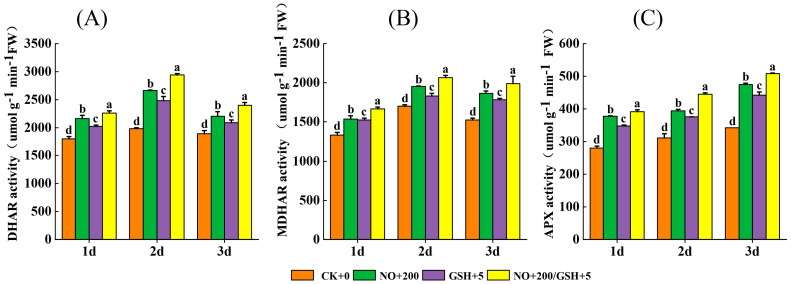
Effects of exogenous NO and GSH on the activities of ((**A**): DHAR), ((**B**): MDHAR), and ((**C**): APX) in leaves of cowpea seedlings under low-temperature stress. Different letters indicate statistically significant differences according to Duncan’s multiple range test at *p* < 0.05.

**Figure 7 plants-12-01317-f007:**
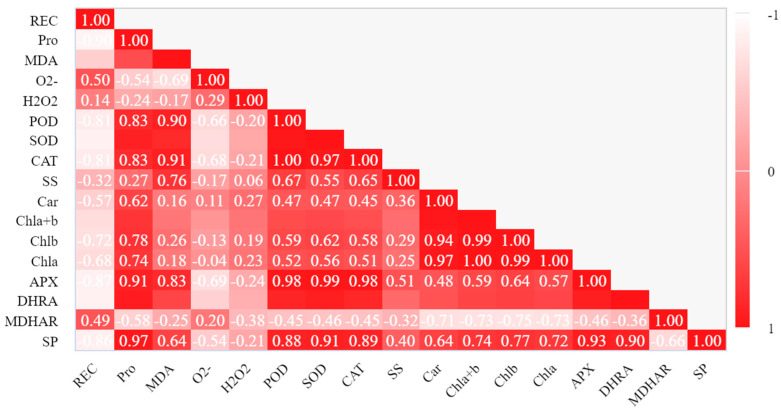
Pearson correlation chart.

**Figure 8 plants-12-01317-f008:**
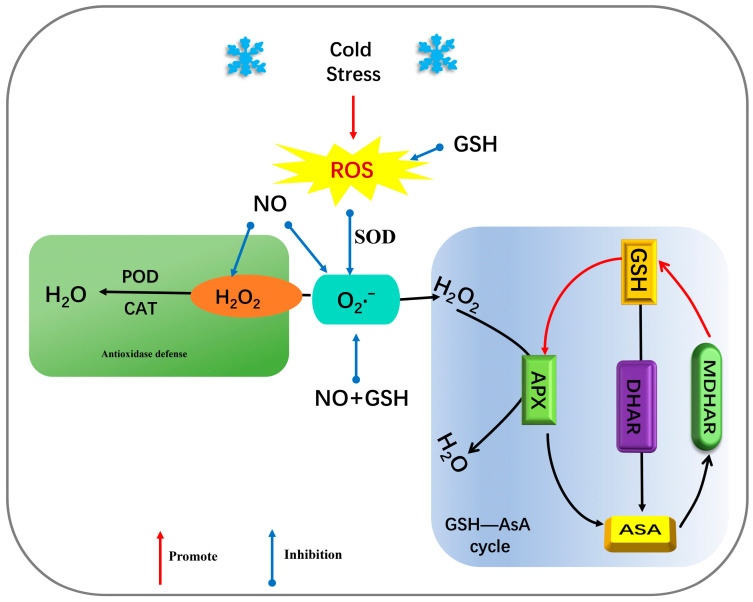
GSH–ASA oxidation system cycle diagram (the red arrow indicates promotion and the blue arrow indicates inhibition).

## Data Availability

Not applicable.

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
