# Peer review of "NO and GSH Alleviate the Inhibition of Low-Temperature Stress on Cowpea Seedlings"

_plants, 2023, doi:10.3390/plants12061317_

Round 1

Reviewer 1 Report

The authors submitted a paper reporting their studies on possible countermeasures to low temperature stress in cowpea seedlings. For this purpose they investigated the effects of nitric oxide and glutathione on the concentrations of ROS, on the activity of enzymes responsible for the detoxification of these highly reactive chemical species, on the degradation of photosynthetic pigments and on the content of malondialdehyde and osmotic regulating substances in cowpea seedling in low temperature conditions.

This kind od studies is bery interesting because they concern the environment, whose balance is increasingly jeopardized by climate change as well as for the effectsin terms of crops yields, food availability and related economy.

In my opinion the paper is noteworthy, but for the pubblication needs a major/minor revision.

Some notes for the Authors:

- Please pay attention in the abstract there is not indication about the low temperature to which the seedlings were subjected. In addition It is not indicated that 200 micromolar is the concentration of NO.

- The introduction needs to be improved , it lacks organicity and some sentences appear very unrelated to each other. There is not reference 12 in the text, only [error!]. What is the reference 12 in the bibliography? It is the right reference? It doesn't seem to be. Moreover, please indicate all the abbreviations at the first citation (GSH-AsA in introduction, but also REC in results section). Only at the first citation (SS, Pro and SP are indicated two times)

- The Results section is before Materials and Methods, in my opinion for a better understanding and an easier reading it woul be good to explain T1, T2 and T3 with a few more words in this previous section. Attention to typos at line 77, that is5mMGSH and at line 80 MAD.

- In the results section, Fig 2 H2O2 content, are correct the letters about the significant differences for days 1 and 3?

- paragraph 2.5 of the results, please explain better these results about enzymatic activities. Now it is difficult to understand the reported activity variations.

- please control all the material and methods section. Please do not use r/m, better rpm or g.

- Line 296 in Materials and Methods section: please better explain the absorbance measures at the three wavelenghts (respectively?)

- Line 316, please change absorb any in absorbance

Author Response

February 3, 2023

Dear Editors and Reviewers:

Thank you for your letter and for the reviewers' comments concerning our manuscript entitled "NO and GSH alleviate the inhibition of low temperature stress on cowpea seedlings "(ID: plants-2175254).   Those comments are all valuable and very helpful for revising and improving our paper, as well as the important guiding significance tour researches.    We have studied comments carefully and have made correction which we hope meet with approval.    Revised portion are marked in red in the paper.    The main corrections in the paper and the responds to the reviewer's comments are as flowing: In the paper and the responds to the reviewer's comments are as flowing:

Reviewer: 1

-1. Please pay attention in the abstract there is not indication about the low temperature to which the seedlings were subjected. In addition, It is not indicated that 200 micromolar is the concentration of NO.

-Response: We are sorry for our omission, and we sincerely thank you for your suggestions in the abstract.  We have added relevant contents to the corresponding places of the manuscript according to your suggestions and marked them in red.

-2. The introduction needs to be improved, it lacks organicity and some sentences appear very unrelated to each other. There is not reference 12 in the text, only [error!]. What is the reference 12 in the bibliography? It is the right reference? It doesn't seem to be. Moreover, please indicate all the abbreviations at the first citation (GSH-AsA in introduction, but also REC in results section). Only at the first citation (SS, Pro and SP are indicated two times)

-Response: Thank you very much for your comments. First of all, regarding the discontinuity of the introduction, we rearrange and revise it. Secondly, we re-checked all references in the manuscript and re-inserted them all in the way of cross-reference. In addition, we also checked all abbreviations in the full text and added the full name where they first appeared, which was added to the corresponding position of the manuscript and marked in red.

-3. The Results section is before Materials and Methods, in my opinion for a better understanding and an easier reading it would be good to explain T1, T2 and T3 with a few more words in this previous section. Attention to typos at line 77, that is5mMGSH and at line 80 MAD.

-Response: Thank you very much for your comments and suggestions. T1, T2 and T3 have been explained in the notes of the first drawing and added in the corresponding positions of the manuscript. The errors in lines 77 and 80 have been corrected and marked in red.

-4. In the results section, Fig 2 H2O2 content, are correct the letters about the significant differences for days 1 and 3?

-Response: We are very sorry for our oversight, and thank you very much for your suggestions on the details of our chart. According to your suggestion, we have added relevant content in the corresponding position of the manuscript and marked it in red.

-5. paragraph 2.5 of the results, please explain better these results about enzymatic activities. Now it is difficult to understand the reported activity variations.

-Response: Thank you very much for your comments. According to your suggestion, we have reviewed the data, reanalyzed all the unclear parts in the drawing and content analysis, and marked the relevant positions in red.

-6. please control all the material and methods section. Please do not use r/m, better rpm or g.

-Response: Thank you very much for your suggestion. We will complete the modification in the corresponding position of the manuscript and mark it in red.

-7. Line 296 in Materials and Methods section: please better explain the absorbance measures at the three wavelenghts (respectively?)

-Response: Malondialdehyde, one of the peroxidation products of plant membrane lipid, reflects the degree of membrane lipid peroxidation, and its content is inversely proportional to the strength of plant stress resistance. MDA reacted with thiobarbituric acid (TBA) under high temperature and acid conditions to form a colored trimethyl complex with maximum light absorption at 532 wavelength and minimum light absorption at 600nm with an absorption coefficient of 155 mmoL/ (L·cm). The concentration of MDA is calculated according to formula (1). However, the carbohydrate in plant tissue interfered with the MDA-TBA reaction. After the error interference caused by sucrose is eliminated by formula (2), the concentration of MDA in the sample extract can be directly obtained. formula (1): C (μ mol/L) = (A532-A600) * 1000/155L, formula (2) C (μ mol/L) =6.45 (A532-A600) - 0.56A450 (A532, A600 and A450 represent the absorbance values at 532nm, 600nm and 450, L is the thickness of the colorimetric cup)

-8. Line 316, please change absorb any in absorbance.

-Response: Thank you very much for your suggestion. We have modified it according to your suggestion and marked it in red.

We tried our best to improve the manuscript and made some changes in the manuscript. These changes will not influence the content and framework of the paper. And here we did not list the changes but marked in red in revised paper. We appreciate for Editors/Reviewers’ warm work earnestly, and hope that the correction will meet with approval.

Once again, thank you very much for your comments and suggestions.

Sincerely,

Song Xueping

College of Horticulture, Sichuan Agricultural University, Chengdu, Sichuan, China

Email address: [email protected]

Corresponding author: [email protected]

Reviewer 2 Report

Dear Authors,

In this study the authors study the effect of NO and GSH on protection against low temperatures in cowpea seedlings. They study numerous different parameters such as the production of superoxide radicals, hydrogen peroxide, MDA, SOD,..... and observe that the spray application of NO and GSH produces an improvement in most of them. Noting that the greatest protection occurs when both are present. The effect of GSH and NO in relieving the effects of different stresses such as cold has already been studied in other varieties of plants, and now the authors are doing it with Cowpea. I consider that the article has a series of style and methodological deficiencies that, except for its correction, make me reject its publication in its current state.

*The biggest critical point that I have to say is that the data at time zero are not shown in any of the conditions studied, and I think this is a fundamental and necessary control. I consider that these data are fundamental for a correct assessment of the results obtained. With the lack of these data, how can we know the effect at 1, 2 and 3 days, if the control of the starting state conditions is lacking? I do not get it. Please, I suggest to the authors that if they have them, they should include these data. And that once they are introduced to the different graphs, they must re-discuss the results based on them.

L285: The true leaves of  two seedlings were cut at 1st day, 2nd day and 3rd day under low temperature stress. That is to say, do you also keep the leaves cut on day zero?

*I have detected numerous syntax and grammatical errors in the use of English. This phrase being just one example of them.  

L11: “In order to study the alleviative effect of exogenous substances nitric oxide (NO) and Glutathione (GSH) on cowpea (Vigna unguiculata (Linn.) Walp.)”  

L356: This?? Under low temperature stress

The article needs a deep review of the use of English.

*Please, In the introduction, refer to the role of NO as a signal molecule in plants.

*Please explain why specifically 200 μmol/L NO and 5 mmol/L GSH were used and not other higher or lower concentrations, in base of  which studies have been fixed?

*There are numerous typographical errors that denote a great lack of care on the part of the authors. I only indicate some of them:

L13: s 200 μmol·L-1 ????? and 5 mmol·L-1 GSH. Please indicate

L 33: (Pro), et al [4].

L49: [Error! Reference source not found.]

L56: can pro-mote

L323: The leaves of 0.2 g were weighed

L316: The absorb any??

L298: 150 r/m. rpm

L177: (Figure 1A B-C.),

L235: Yao [18] et al.

*Only in Fig. 1 the panels are differentiated as A,B,C. It would be convenient that in the rest of the figures were also  indicated A,B…, this would improve the reference to them in the text.

*Figure 4, In the two graphs above the Y axis legend is the same, I guess one of them is actually Chlorophyll a

*The references need a thorough revision, there are numerous errors that can be seen with the naked eye.

*The legend of figure 7 must be completed. Indicating that the data corresponds to the p-values. replace relative conductivity with REC.

* The legend of Fig. 8 is absent, and it must be self-explanatory, so as not to go to the text to understand what they mean, it must be indicated what the authors mean in the legend too.

L319: The content of soluble sugar (SS) was determined by an-throne?? colorimetric method. What method are you referring to? indicate the reference, please.

* L179: “Some studies suggest that NO may act on H2O2 in mediating stress tolerance [34] (Figure 8)” Please indicate what those students say, the figure does not explain anything

L193: “Exogenous NO significantly increased the endogenous NO content” Logically, otherwise this paper would make little sense, but cite this fact correctly, please. But I think you will refer to NO donors as the SNP, right?

L196: “seedlings under low temperatures [38]” I don't understand what this reference contributes, can you explain it better, please?

L206: “It is possible that NO mediated chlorophyllase and Mg dehydrogenase activities, limiting the degradation of chlorophyll or increasing the content of Mg, an element necessary for chlorophyll synthesis, thereby accelerating the synthesis of plant photosynthetic pigments [40]??” In the article cited, no reference is made to the NO

L224: “which proved that plants could increase their tolerance to low temperature through the antioxidant enzyme system”. This was known prior to this study. In any case, cite the appropriate reference

L227: “MAPK pathway” cite the correct paper

L239: “positive correlated ( p<0.01) with the malondialdehyde (Figure 8 7.)

Author Response

February 3, 2023

Dear Editors and Reviewers:

Thank you for your letter and for the reviewers' comments concerning our manuscript entitled "NO and GSH alleviate the inhibition of low temperature stress on cowpea seedlings "(ID: plants-2175254).   Those comments are all valuable and very helpful for revising and improving our paper, as well as the important guiding significance tour researches.    We have studied comments carefully and have made correction which we hope meet with approval.    Revised portion are marked in red in the paper.    The main corrections in the paper and the responds to the reviewer's comments are as flowing: In the paper and the responds to the reviewer's comments are as flowing:

Reviewer: 2

  1. *The biggest critical point that I have to say is that the data at time zero are not shown in any of the conditions studied, and I think this is a fundamental and necessary control. I consider that these data are fundamental for a correct assessment of the results obtained. With the lack of these data, how can we know the effect at 1, 2 and 3 days, if the control of the starting state conditions is lacking? I do not get it. Please, I suggest to the authors that if they have them, they should include these data. And that once they are introduced to the different graphs, they must re-discuss the results based on them.

-Response: Thank you very much for your comments and suggestions. We found from the practice and reference of the relevant literature as follows [][] that when normally growing plants are subjected to low temperature stress, the related physiological changes will appear the same. In addition, we can also observe this change in the phenotypic diagram. Therefore, according to our summary above, our experiment starts directly from the first day of low temperature with sampling and measurement.

L285: The true leaves of two seedlings were cut at 1st day, 2nd day and 3rd day under low temperature stress. That is to say, do you also keep the leaves cut on day zero?

-Response: Thank you very much for your comments. On day 0, the seedling leaves were all present, that is, the seedling leaves in normal growth without exogenous application before low temperature treatment.

2.* I have detected numerous syntax and grammatical errors in the use of English. This phrase being just one example of them.  

L11: “In order to study the alleviative effect of exogenous substances nitric oxide (NO) and Glutathione (GSH) on cowpea (Vigna unguiculata (Linn.) Walp.)”  

L356: This?? Under low temperature stress

The article needs a deep review of the use of English.

-Response: We are very sorry to bring you a bad reading experience due to our poor English level, but we have carefully revised and proofread the manuscript and marked it in red.

3.*Please, In the introduction, refer to the role of NO as a signal molecule in plants.

-Response: Thank you very much for your comments. According to your suggestion, we have added the role of NO as a signaling molecule in relevant positions and marked it in red.

4.*Please explain why specifically 200 μmol/L NO and 5 mmol/L GSH were used and not other higher or lower concentrations, in base of which studies have been fixed?

-Response: Thank you for your comments and questions. This study is based on literature review and previous experimental results, such as reference [1][2]. After low temperature stress, NO was sprayed at 200 μmol/L, GSH sprayed at 5 mmol/L has the best alleviating effect. Therefore, this concentration is also used for the experiment.

[1] Wu P,  Xiao C ,  Cui J , et al. Nitric Oxide and Its Interaction with Hydrogen Peroxide Enhance Plant Tolerance to Low Temperatures by Improving the Efficiency of the Calvin Cycle and the Ascorbate–Glutathione Cycle in Cucumber Seedlings[J]. Journal of Plant Growth Regulation, 2020:1-19.

[2] Md Tahjib-Ul-Arif , Xiangying Wei, Israt Jahan, et al. Exogenous nitric oxide promotes salinity tolerance in plants: A meta-analysis[J]. Frontiers in Plant Science (IF 6.627 ) Pub Date : 2022-11-07 , DOI: 10.3389/fpls.2022.957735.

4.*There are numerous typographical errors that denote a great lack of care on the part of the authors. I only indicate some of them:

L13: s 200 μmol·L-1 ????? and 5 mmol·L-1 GSH. Please indicate

L 33: (Pro), et al [4]. L49: [Error! Reference source not found.]

L56: can pro-mote

L323: The leaves of 0.2 g were weighed

L316: The absorb any??

L298: 150 r/m. rpm

L177: (Figure 1A B-C.),

L235: Yao [18] et al.

-Response: We are very sorry for our oversight, and thank you very much for your suggestions on the details of our chart. According to your suggestion, we have added relevant content in the corresponding position of the manuscript and marked it in red.

5.*Only in Fig. 1 the panels are differentiated as A, B, C. It would be convenient that in the rest of the figures were also indicated A, B…, this would improve the reference to them in the text.

-Response: Thank you very much for your suggestion. We have modified all the pictures in the manuscript and inserted them in the corresponding positions with red marks.

6.*Figure 4, In the two graphs above the Y axis legend is the same, I guess one of them is actually Chlorophyll a.

-Response: We are deeply sorry for our carelessness and mistakes. Thank you very much for your suggestions. We have completed all modifications and marked them in red.

7.*The references need a thorough revision, there are numerous errors that can be seen with the naked eye.

-Response: Thank you very much for your valuable suggestions. We have completed the modification according to your suggestions and marked it in red.

8.*The legend of figure 7 must be completed. Indicating that the data corresponds to the p-values. replace relative conductivity with REC.

-Response: Thank you very much for your opinion. Figure 7 has been modified according to your suggestions and marked in red in the manuscript.

* The legend of Fig. 8 is absent, and it must be self-explanatory, so as not to go to the text to understand what they mean, it must be indicated what the authors mean in the legend too.

-Response: Thank you very much for your opinion. FIG. 8 describes the changes of related enzyme activities of GSH-ASA system in seedlings after NO+GSH was applied to alleviate low temperature stress in seedlings. Red arrows indicate promotion and blue arrows indicate inhibition, which have been marked in red at relevant locations in the manuscript.

9.L319: The content of soluble sugar (SS) was determined by an-throne?? colorimetric method. What method are you referring to? indicate the reference, please.

-Response: Thank you very much for your comments. The sugar content was determined using anthrone colorimetric method. The reference is 28, which has been supplemented in the relevant places of the manuscript and marked in red.

10.* L179: “Some studies suggest that NO may act on H2O2 in mediating stress tolerance [34] (Figure 8)” Please indicate what those students say, the figure does not explain anything

-Response: Thank you very much for your suggestions and questions. This study showed that spraying exogenous NO+GSH on seedlings subjected to low temperature stress could effectively reduce the content of H2O2, thus reducing the damage caused by low temperature stress. Similar results were also obtained from other plants, as shown in the reference [32][33][34]. Therefore, the possible mediated relationship between NO and H2O2 after low temperature stress was indicated by the flow chart of related enzymes in GSH-ASA system.

L193: “Exogenous NO significantly increased the endogenous NO content” Logically, otherwise this paper would make little sense, but cite this fact correctly, please. But I think you will refer to NO donors as the SNP, right?

L196: “seedlings under low temperatures [38]” I don't understand what this reference contributes, can you explain it better, please?

-Response: Thank you very much for your valuable comments on my manuscript. The references are speculative conclusions put forward according to our experimental results. Maybe there are some deviations, but now we have deleted the references that are inconsistent with the experimental results.

L227: “MAPK pathway” cite the correct paper

-Response: Thank you very much for your comments. We reinserted the correct references at the appropriate places in the manuscript and cross-referenced them in the order in our manuscript.

L239: “positive correlated (p<0.01) with the malondialdehyde (Figure 8 7.)

-Response: Thank you very much for your comments. We are rechecking the parts that have been redacted.

We tried our best to improve the manuscript and made some changes in the manuscript. These changes will not influence the content and framework of the paper. And here we did not list the changes but marked in red in revised paper. We appreciate for Editors/Reviewers’ warm work earnestly, and hope that the correction will meet with approval.

Once again, thank you very much for your comments and suggestions.

Sincerely,

Song Xueping

College of Horticulture, Sichuan Agricultural University, Chengdu, Sichuan, China

Email address: [email protected]

Corresponding author: [email protected]

Reviewer 3 Report

Dear Authors

Your work dealing with NO and GSH treatments to cowpea seedlings is based on a good experimental idea but I am very sorry to say that in my opinion is not suitable for publication in this form.

First of all the English must be revised thoroughly because sometime is not clear the meaning of your sentences. For instance, row 24: plant geographical analys. Do you mean the geographical distribution of the plants? Row 268: what do you mean for soup? Was it water or something else? Row 270: were exposed to white. I cannot understand the meaning.

Also the precision of our exposition must be revised what is a: deep hole disc (4 x 8 size) row 271. Are cm or inches, fabric, soil, glass, I cannot understand. And at row 273: the soil is either dry or wet. You MUST be precise to give information to repeat the experiment. Your description of the experiment is imprecise. E.g. Row 279 each treatment was repeated three times: when, how?

When programming an experiment like your, usually there is a point 0 then day 1, day 2 day 3 and so forth. You did not include a starting point for your analysis, and I think this is a drawback because you do not have the original initial condition of the plants. In my opinion this would be enough to reject your paper. As a matter of fact even without the initial point your results could be of some interest but the presentation and the statistics is not correctly presented. Let’s start from the simplest case: row 157 you write there was no difference between…. This is completely wrong. You are dealing with correlations among variables. You should show a 17 x 17 heat chart with the names of all the variables and then the sentence in the text should be related to the presence or absence of correlation.

Since you have four treatments and time as variable in the system you must also show the statistics and significance not only within each day but also among the days. In your graphs (let’s take the figure 2) I can see the content of H2O2 but I cannot understand if the level of H2O2 in the NO+200 differs statistically among the three dates. You should use the ANOVA procedure and them show the ANOVA table then maybe place a graph to better explain your findings.

Also in the text there are so many errors I cannot write all of them: two b letters in fugure 2 H2O2 1d, a t lacking in ordinate text of the figure 3, the CK+0 is not clear it is just CK or Control (C or CT) a reference missing at row 49.

Finally you should check the instruction for authors. The introduction should “briefly mention the main aim of the work and highlight the main conclusions. Keep the introduction comprehensible to scientists working outside the topic of the paper”. It is impossible to understand your aim and the main results of your work. 

Author Response

February 3, 2023

Dear Editors and Reviewers:

Thank you for your letter and for the reviewers' comments concerning our manuscript entitled "NO and GSH alleviate the inhibition of low temperature stress on cowpea seedlings "(ID: plants-2175254).   Those comments are all valuable and very helpful for revising and improving our paper, as well as the important guiding significance tour researches.    We have studied comments carefully and have made correction which we hope meet with approval.    Revised portion are marked in red in the paper.    The main corrections in the paper and the responds to the reviewer's comments are as flowing: In the paper and the responds to the reviewer's comments are as flowing:

Reviewer: 3

Your work dealing with NO and GSH treatments to cowpea seedlings is based on a good experimental idea but I am very sorry to say that in my opinion is not suitable for publication in this form.

1.First of all the English must be revised thoroughly because sometime is not clear the meaning of your sentences. For instance, row 24: plant geographical analys. Do you mean the geographical distribution of the plants? Row 268: what do you mean for soup? Was it water or something else? Row 270: were exposed to white. I cannot understand the meaning.

-Response: Thank you very much for your opinion. The sentence Row24 "plant geographical analys." in the manuscript means that high temperature or low temperature has certain influence on the geographical distribution of plants. Row268: soup means soaking seeds in warm water at 30-55℃. Row 270: "were exposed to white" means that cowpea seeds with the same growth were planted in the hole tray after their white tips had sprouted. The above questions are revised and marked in red where relevant in the manuscript.

2.Also the precision of our exposition must be revised what is a: deep hole disc (4 x 8 size) row 271. Are cm or inches, fabric, soil, glass, I cannot understand. And at row 273: the soil is either dry or wet. You MUST be precise to give information to repeat the experiment. Your description of the experiment is imprecise. E.g. Row 279 each treatment was repeated three times: when, how?

-Response: Thank you very much for your comments and suggestions. First of all, each hole of the hole plate we use to raise seedlings is (6 cm long ×11 cm high×6 cm wide), and a hole plate has a total of 32 holes. Nutrient soil for seedling cultivation: perlite: fine coconut husk: peat soil for 2:3:5. Second, "row 273: he soil is either dry or wet." That is, when the nutrient soil moisture is 60%.

Finally, Row 279: We are very sorry that your reading may have been obstructed by the improper organization of our sentences. We revised and supplemented the manuscript and marked it in red and green. Specific modifications are as follows: 12-13 days later, according to the growth of the second pair of true leaves, the first pair of true leaves fully unfolded),  select the seedlings with good growth and consistent development state for exogenous treatment.  Before low temperature   treatment, cowpea seedlings were placed in artificial intelligence climate box for preculture. The pre-culture conditions were temperature 25/18℃, photcycle 12h/12h (day and night), light intensity 300 μmol m-2 s-1, and relative humidity 75%. After 24 h of preculture, low temperature treatment was carried out at 8/8℃, and relative humidity 75%. After 24 h of preculture, low temperature treatment was carried out at 8/8℃, light intensity of 300 μmol m-2 s-1 and relative humidity of 75%. It could be divided into four treatments, namely clear water control (CK), 200 μmol/L NO (T1), 5 mmol/L GSH (T2) and 200 μmol/L NO+5 mmol/L GSH (T3). Water droplets did not drop until the surface of the leaves was sprayed. There were 16 plants per treatment, and each treatment was repeated 3 times and simultaneously.

3.When programming an experiment like your, usually there is a point 0 then day 1, day 2 day 3 and so forth. You did not include a starting point for your analysis, and I think this is a drawback because you do not have the original initial condition of the plants. In my opinion this would be enough to reject your paper. As a matter of fact, even without the initial point your results could be of some interest but the presentation and the statistics is not correctly presented. Let’s start from the simplest case: row 157 you write there was no difference between…. This is completely wrong. You are dealing with correlations among variables. You should show a 17 x 17 heat chart with the names of all the variables and then the sentence in the text should be related to the presence or absence of correlation.

-Response: Thank you very much for your comments and suggestions. We found from the practice and reference of the relevant literature as follows [1], [2] that when normally growing plants are subjected to low temperature stress, the related physiological changes will appear the same. In addition, we can also observe this change in the phenotypic diagram. Therefore, according to our summary above, our experiment starts directly from the first day of low temperature with sampling and measurement. Secondly, as for the correlation analysis, we re-analyzed and revised it in English, and marked it with red revision in the relevant position of the manuscript.

[1] Hussain H A,  Saddam H ,  Abdul K , et al. Chilling and Drought Stresses in Crop Plants: Implications, Cross Talk, and Potential Management Opportunities[J]. Frontiers in Plant Science, 2018, 9:393-.

[2] Yingying Zhang, Taoyu Dai, Yahui Liu, et al. Effect of Exogenous Glycine Betaine on the Germination of Tomato Seeds under Cold Stress[J]. International Journal of Molecular Sciences ( IF 4.556 ) Pub Date : 2022-09-09 , DOI: 10.3390/ijms231810474

4.Since you have four treatments and time as variable in the system you must also show the statistics and significance not only within each day but also among the days. In your graphs (let’s take the figure 2) I can see the content of H2O2 but I cannot understand if the level of H2O2 in the NO+200 differs statistically among the three dates. You should use the ANOVA procedure and them show the ANOVA table then maybe place a graph to better explain your findings.

-Response: Thank you very much for your suggestions. According to your suggestions on the manuscript, we have made the variance analysis of time variables and related indicators, and added these contents as attachments in the corresponding positions of the manuscript.

5.Also in the text there are so many errors I cannot write all of them: two b letters in figure 2 H2O2 1d, a t lacking in ordinate text of the figure 3, the CK+0 is not clear it is just CK or Control (C or CT) a reference missing at row 49.

-Response: We are very sorry for the loopholes in our work and sincerely thank you for your questions and valuable comments. We revised the drawings and references and marked them in red at the corresponding positions of the manuscript. CK+0 stands for low temperature control.

6.Finally you should check the instruction for authors. The introduction should “briefly mention the main aim of the work and highlight the main conclusions. Keep the introduction comprehensible to scientists working outside the topic of the paper”. It is impossible to understand your aim and the main results of your work.

-Response: Thank you very much for your advice. We rewrote the introduction and revised the content and syntax. We also checked the author's notes and highlighted them in red.

We tried our best to improve the manuscript and made some changes to the manuscript. These changes will not affect the content and framework of the paper. Instead of listing these changes here, we have highlighted them in red in the revised paper. In addition, as for the variance analysis between different variables you proposed, due to time, we are completing the chart, which will be submitted to the link you provided on February 4, 2023. We sincerely thank the editors/reviewers for their enthusiastic work and hope that the revision will be approved.

Thank you again for your comments and suggestions.

Sincerely,

Song Xueping

College of Horticulture, Sichuan Agricultural University, Chengdu, Sichuan, China

Email address: [email protected]

Corresponding author: [email protected]

Round 2

Reviewer 2 Report

Dear Authors,

I believe that the authors have responded favorably to most of my comments and I accept the paper in its current version.

Reviewer 3 Report

Dear Authors,

thank you for your work on the paper which was improved thoroughly. Thank you also for accepting my suggestion. I am not native English speaking but the vocabulary is useful:

Analysis= separation, partition, subdivision

Distribution= frequency, arrangement, occurrence, pattern, number, population, spread, concentration.

Please change “plant geographical analysis" with “plant geographical distribution” or with a synonym of distribution because your sentence is not clear.

Please change the complicate sentence in all your graphs:

The differences among the treatments indicated with the same letter vertically were not significant according to Duncan’s multiple range test at P < 0.05.

To a much simpler one:

Different letters indicate statistically significant differences according to Duncan’s multiple range test (P≤0.05).